# Pro-Environmental Transformation of the Equine Sector—Facilitators and Challenges

**DOI:** 10.3390/ani14060915

**Published:** 2024-03-16

**Authors:** Susanna Hedenborg, Mathilde Kronborg, Anna Sätre, Aage Radmann, Gabriella Torell Palmquist, Petra Andersson

**Affiliations:** 1Department of Sport Science, Malmö University, 205 06 Malmö, Sweden; susanna.hedenborg@mau.se; 2Department of Teacher Education and Outdoor Studies, Norwegian School of Sport Sciences, 0863 Oslo, Norway; mathilde.kronborg@nih.no (M.K.); aager@nih.no (A.R.); 3Strömsholm Equestrian Centre, 734 94 Strömsholm, Sweden; gabriella.torell-palmquist@rsflyinge.se; 4Department of Educational Studies, Karlstad University, 651 88 Karlstad, Sweden; 5Department of Philosophy, Linguistics and Theory of Science, University of Gothenburg, 405 30 Göteborg, Sweden; petra.andersson@filosofi.gu.se

**Keywords:** equine sector, environmental challenges, COM-B model, horse welfare

## Abstract

**Simple Summary:**

To improve horse welfare and ensure a sustainable equestrian future, we need to understand human behavior in relation to the challenges of the sector. This article maps and analyzes how individuals within the equine sector in Sweden and Norway define environmental challenges. An analysis based on a model for behavior change (the COM-B model) shows that there is a capacity for change, and that skills and knowledge exist, but that some individuals desire more information and a deeper understanding of the issues at hand. Physical constraints such as location, resources, and time seem challenging for individual actors to influence. Economic factors are also identified as impediments to transformation. Cultural norms related to orderliness within stables, although not directly addressing ecological challenges, might serve as a foundation for promoting environmental initiatives that will also improve horse welfare.

**Abstract:**

To improve horse welfare and ensure a sustainable equestrian future, we need to understand human behavior in relation to the challenges of the sector. The purpose of this paper is to map and analyze how individuals within the equine sector in Sweden and Norway define the environmental challenges they are faced with and how these are related to questions about horse welfare. A mixed-methods parallel design was used. The data consist of survey answers and semi-structured interviews. The survey, responded to by 697 Swedish and Norwegian participants, ensured statistical validity and power through a sample size calculation yielding approximately 385 participants. To deepen the understanding, 36 semi structured interviews with Swedish and Norwegian interviewees were conducted. An analysis of convergencies and divergencies between the data sets provided robust insights into the perceptions and behaviors within the equine sector in Sweden and Norway. The findings show that the equine sector has cultivated a stronger environmental commitment over the last 15 years (Svala, 2008). However, many participants express a perceived lack of influence on this transformation. The COM-B model (Michie, Van Stralen & West, 2011; Michie, Atkins & West, 2014) and previous research on ‘thinking structures on climate delay’ (Wormbs & Wolrath, 2023) are used to interpret the data. The analysis indicates that there is an overall capacity for change, and that skills and knowledge exist, but some individuals desire more information and a deeper understanding of the issues at hand. Higher barriers to change were found in the opportunity component, where physical constraints such as location, resources, and time seem challenging for individual actors to influence. Economic factors are also identified as impediments to transformation. Cultural norms related to orderliness within stables, although not directly addressing ecological nor ethical challenges, might serve as a foundation for promoting environmental initiatives that will also improve horse welfare.

## 1. Introduction

The equine sectors in Sweden and Norway provide recreational activities, entertainment, and employment for individuals of all ages. The Swedish equine sector is larger than the Norwegian one. In 2019, Sweden had approximately 355,000 horses (3.55 horses per 100 persons), generating a turnover of 32 billion SEK and creating 18,500 jobs [1]. The Norwegian statistics are not up to date, but a government report from 2018 highlighted the existence of 125,000 horses in 2012 (2.5 horses per 100 persons). The sector contributes significantly to the economy in Norway too [2]. While the equine sector offers positive societal effects such as outdoor activities and recreation, it also poses environmental challenges [3,4,5]. Journalist Arne Müller [6] characterizes the Swedish equine sector as a ‘climate villain’, emitting more carbon dioxide than domestic air travel. Müller concludes that, even with limited research, feed production and intra-sector transport pose severe environmental threats. Scientific studies on the equine sector arrive at similar conclusions, highlighting environmental issues such as nutrient leakage from manure piles, pastures, and grazing fields [7].

In response to the questioning of the equine sector’s climate footprint, voices from the sector have shown an awareness of the challenges and are involved in ongoing efforts to counteract negative environmental impacts. Publications on horses and sustainable development [2], along with policy documents and local recommendations for equine activities, have been created to inform and inspire the sector [8,9,10,11]. However, little is known about how recommendations and policies are operationalized in practice. This dearth of empirical literature necessitates studies that evaluate the strategies used to implement environmental practices.

Research indicates that sector-wide transformations are challenging [12] because knowledge of environmental impact alone is insufficient to change a sector. There is a need to understand how people define and reason in relation to pro-environmental behavior changes to build structures that support pro-environmental transformation in many areas. A review by Wolframm and her colleagues points to the importance of behavior change in the equine sector for equestrianism to flourish [13]. They work with the COM-B model and the Behavior Change Wheel (BCW) and focus on ethics and equine wellbeing. In this article, we work with these models too and argue that the equine sector provides an interesting case as a pro-environmental transformation must be carried out without compromising horse welfare to ensure a sustainable equestrian future. The purpose of this article is to map and analyze how individuals within the equine sector in Sweden and Norway define the environmental challenges they are faced with and how these are related to questions about horse welfare.

## 2. Environmental Research within Sport Science and the Equine Sector

The environmental research within sports science has predominantly focused on the impact of factors such as high altitude, heat, cold, and air pollution on the human body [14,15,16]. There is, however, a growing number of studies within sustainable sport management and sport ecology [17]. Often, studies within the field of Sport Management have explored the environmental effects of events and mega-events [18,19,20]. However, studying the effects is not enough and Brian McCullough [17] argues that researchers must study how sport organizations respond to climate change. An example of this is a study by Daniel Svensson et al. [21], which explores the carbon footprint and environmental challenges of sports based on Allen Guttmann’s sportification model. Guttmann’s model illustrates the transformation of sports from rituals to record-focused activities, encapsulated by the notion of ‘faster, higher, stronger’, which, according to the authors, does not straightforwardly align with active environmental efforts. Despite this, Svensson et al. demonstrate that sports entities have long been working with various regulatory models, such as anti-doping measures and prohibitions on certain equipment and styles for jumping, running, and throwing, which seem rather constraining with regard to athletes achieving new records. The study suggests that these regulatory models could potentially address environmental challenges in a similar manner. Another example is a study by Anne Tjønndal et al. [22], which examines sustainability efforts within Norwegian sports and outdoor activities, revealing three overarching tensions: ecological and economic, social and ecological sustainability in practice, and social sustainability and the competitive focus of sports. In Sweden, Marie Larneby et al. show both strengths and weaknesses in ongoing environmental efforts within sports and outdoor organizations [23,24]. The studies indicate conflicting sustainability goals and challenges in implementing environmental initiatives within member organizations where participation is motivated by factors other than an interest in environmental issues. Despite resource constraints, National Sports Federations (NSFs) in Sweden view environmental issues as significant and have created several pro-environmental policies. While there is a lack of resources for environmental projects, cross-federation collaboration has been proposed as one avenue to advance efforts. Ecological sustainability with regard to transportation is identified as a major challenge. Larneby et al. suggest that policy goals, particularly for transportation, often revolve around reducing carbon dioxide emissions rather than promoting learning, such as understanding the environmental impact of specific sports activities or events [23,24]. Equestrian sports face similar challenges as other sports, not least in relation to questions of how human behavior can be influenced and changed. There are, however, additional challenges in relation to the human use of animals. In the equine sector (and sports), steps towards a more sustainable future must be taken with a starting point in ensuring (and possibly improving) horse welfare. Equestrian sports depend on societies’ acceptance of human–horse practices and horse welfare is a cornerstone for the equine sectors’ social license to operate (SLO) [25,26,27]. There are, however, other practices related to social and ecological sustainability creating acceptance too. Both the Norwegian and Swedish Equestrian Federations underline that they need to work with all sustainability goals to ensure SLO. In this study, we will argue that several pro-environmental changes in the sector can work in favor of such development.

In studies of negative environmental impacts caused by horse keeping, nutrient leakage (such as nitrogen and phosphorus) from manure piles, pastures, and grazing fields has received particular attention [28,29]. Another theme concerns rapid and ongoing changes in landscape use, especially in countries like Sweden and other medium- to high-income nations, where small farms have ceased operations due to structural changes in agriculture. These small farms are now often repurposed into ‘horse farms’ for equine activities instead [30,31,32]. These changes have led to unforeseen conflicts and cultural clashes [31], such as, for example, how to manage horse-related activities near people who are involved in other leisure practices, which might require considering allergens and ski tracks. Municipalities have been slow to react to this kind of development, and their documents and policies reflect reactions rather than strategies [33]. Horses have often been viewed as problems in the neighborhood that must be solved by the equine community themselves rather than as an asset that can benefit everyone and a sustainable future. An exception is Palmgren [34], who studied horses as ‘landscape caretakers’, emphasizing how grazing and deworming (eliminating intestinal parasites, such as worms, using medication) positively impact local biodiversity. The horses need to graze and the provision for this points to how horse welfare and ecological sustainability can go hand in hand.

There is a lack of systematically conducted empirical studies on how stakeholders in the equine sector perceive challenges and solutions related to environmental policies and local practices. Svala’s study from 2008 is an exception, as it reveals a low interest in sustainable development in the equestrian community and a general skepticism towards ‘ecological things’ [35]. To some extent, this skepticism was based on ideas of horse (and human) welfare related to safety issues being of higher importance than for example pro-environmental issues. However, as her study is more than a decade old, values, perceptions, and ideas about environmental issues and their relationship to horse welfare may have developed. To be able to nuance the discussion of arguments, we relate our results to Wormbs and Wolrath Söderberg’s study of how people reason around their behavior. In their study, they pinpoint people who want to contribute to a transition towards pro-environmental change (rather than people who are indifferent) to demonstrate that even pro-environmentally motivated people present arguments in which paradoxes and challenges against changing their behavior in a more pro-environmental direction are found [36]. They show that a common argument is related to an imagined climate account where a person, through different behaviors, can keep to a budget. Another way of arguing is based on comparing one’s own behavior to something or someone that is worse. Other arguments evolve around redirecting responsibility, the limits of reality, goal conflicts, and the human condition. Whether, and in that case how, these ways of arguing appear in our material will be discussed.

## 3. Theoretical Framework

In this study, we will, through a mixed method parallel design, present how individuals in the equine sector perceive pro-environmental challenges. For this purpose, we have chosen the COM-B model as our theoretical framework. The COM-B model is a model for behavioral change developed by the British health psychologist Susan Michie. It is behavior-oriented and aims to clarify how behaviors are governed and can be altered [37,38]. The COM-B model amalgamates 26 other models of behavior change and is part of a larger framework known as the Behavior Change Wheel (BCW). The BCW is built upon 19 other frameworks and can be used to plan an intervention. Michie et al. argue that interventions are often designed without a formal analysis of either the target behavior or the theoretically predicted mechanisms. Instead, interventions are frequently based on implicit assumptions about behaviors, such as spreading knowledge, even though barriers to change may involve factors other than ignorance, such as lack of infrastructure or motivation. The COM-B model allows for more precise preparation for an intervention.

The letters in the COM-B model stand for the components: behaviors (B) influenced by our capabilities (C), opportunities (O), and motivation (M) [37,38]. Each of these three main components comprise various subparts. To execute any action, a person (or organization) needs to have the capability to change. Capability includes necessary knowledge (referred to as psychological capability) and skills (physical capability). Psychological capability encompasses having the required knowledge to perform a behavior, as well as mental skills such as attention, memory, or decision-making abilities. Physical capability involves bodily capacities, such as possessing the strength, endurance, or skill required for a behavior. The person (or organization) must also have both physical and social opportunities to change. Physical opportunities refer to elements such as location, resources, time, and physical obstacles, while social opportunities involve relationships and cultural norms. Additionally, the person (or organization) must have the motivation to change their behavior. The COM-B model distinguishes between automatic motivation and reflective motivation. Automatic motivation includes habits, natural drives, instincts, desires, needs, and impulses and reflex responses, while reflective motivation involves planning, evaluation, and values. These components influence each other, and a change in one can lead to a change in another. Although Michie et al. have primarily worked on changing various health-related behaviors, the model, focusing on behavior change, can be applied to other areas. To illustrate this point, consider an example related to promoting cycling to reduce carbon dioxide emissions. Applying our hypothetical example to the COM-B model, individuals targeted for taking-up or increasing their adoption of cycling need knowledge about the environmental benefits of cycling (psychological capability) and the ability to cycle (physical capability). Furthermore, there should be cycling lanes (physical opportunities). Additionally, friends, colleagues, or family should support or at least not oppose cycling (social opportunities). Individuals must also be motivated to cycle (motivation). Regarding the COM-B’s third component of motivation, in our hypothetical scenario, individuals might have an automatic motivation to use their cars. What might be needed, however, is a more reflective motivation, where individuals reflect on both their motivation and the information provided through the other two components. Depending on the missing capability, opportunity, or motivation (perhaps lacking the skill to cycle, well-lit cycling lanes, or the desire), an intervention can then be implemented, such as cycling to school, teaching individuals to cycle, expanding cycling lanes, or having motivational conversations. Following the COM-B framework, only through addressing the three components can a change in behavior be expected.

In this article, we will first describe how individuals involved in the equine sector in Sweden and Norway depict the environmental challenges. Subsequently, based on the components of the COM-B model, we will identify capabilities, opportunities, and motivations for undertaking pro-environmental initiatives. The COM-B model implies that we use a simplified understanding of how individuals think and act. For instance, individuals may have multiple, perhaps even conflicting, motives for an action, and there may be dissonance between how a person would like to act and how that person acts [39]. Research shows how cognitive dissonance arises when we act in a way that contradicts our self-image [40,41,42]. Our findings do not provide answers as to whether respondents experience cognitive dissonance based on how they perceive themselves in relation to the environment and sustainable development and how they live and act. Despite this element not being explored, the COM-B model and our findings provide us with an opportunity to begin unraveling how work centered around change can be conducted in the equine sector.

## 4. Methods and Data

We used a mixed-methods design with a parallel convergent approach [43], and the data consist of responses to our online survey and semi-structured interviews.

### 4.1. Survey

The survey was published on 8 December 2022 and left open until 5 June 2023. A link to the survey was distributed through the Swedish Equestrian Federation and the Norwegian Equestrian Federation. It was shared on social media platforms (Facebook) and sent to riding schools that had expressed interest in participating in the study. Exclusion criteria were age (below 16 years) and relation to the equine sector (not active in the sector). The survey received responses from a total of 697 individuals, with 418 providing answers to all questions. We have utilized all responses and specified the number of respondents for each question in the tables and figures below. We determined the sample size for our study to achieve a 95% confidence level and a 5% margin of error using the formula: [z2 × p(1 − p)]/e2/1 + [z2 × p(1 − p)]/e2 × N], with parameters: Confidence Level: 95% (Z-score of 1.96); Margin of Error: 5% (0.05); Estimated Population Proportion: 50% (0.5). Substituting and computing, we obtained a sample size of approximately 385 participants. This calculation ensures statistical validity and power in our study.

The majority of survey respondents were women (96%). This is not surprising, as equestrian sports are numerically dominated by women and girls in Sweden and Norway. In 2022, 93% of the Swedish Equestrian Federation’s members (154,130 individuals) were women and girls [44]. A similar pattern is observed in Norway. In 2021, 88.5% of the Norwegian Equestrian Federation’s members (29,159 individuals) were women and girls [45]. Those who responded to the survey can be described as highly active in the equine sector. More than half of the respondents (54%) answered that they go to the stable every day, and when they go, most spend between two and four hours there (75%).

To gain a deeper understanding of those who responded, we asked people to indicate whether they were hobby riders, horse owners, riding school riders, professionals (such as managers, drivers, stable workers, trainers, veterinarians, and farriers), and whether they were part of the organization’s board. It was possible to choose more than one category. We then grouped respondents into the following categories: have horses as a hobby (including riding school riders, horse owners, and hobby riders), work in the sector (including managers, drivers, stable workers, trainers, veterinarians, farriers), and others (those who selected ‘others’ and those who indicated that they are board members). There were also individuals who selected both having horses as a hobby and being part of the ‘work in the sector’ group—for them, we created a separate category. Table 1 below describes the population. There are more Swedish responses than Norwegian—about two-thirds of the responses come from Sweden (71%) and one-third comes from Norway (29%). The table also shows the mean age for each group and the mean age of those who responded is 41 years in Sweden and 29 years in Norway. The respondents have been in the equine sector for 29 years (Sweden) and 19 years (Norway).

In addition to background information about respondents’ gender, age, and experience in equestrian sports, we asked questions about whether they consider environmental issues important for the equine sector, which environmental challenges they find most significant, the degree to which they believe they have the opportunity and desire to influence environmental work, how they think they can change their behavior to act in a more pro-environmentally friendly way, what obstacles they see for such a change, and what factors they think could motivate someone to change their behavior. The survey featured multiple-choice questions along with opportunities to elaborate on their answers through free text.

### 4.2. Interviews

In addition to survey answers, the study is based on 35 semi-structured interviews. The interviews were conducted from August to December 2022, in both Norway and Sweden. Interviews were conducted by one or two researchers in the team. When interviews were conducted by one researcher, this person had extensive experiences in doing qualitative interviews and the equine sector. When interviews were conducted by two researchers, we made sure that the researchers were matched so that one of them had extensive experience of the method and the other was familiar with the equine sector or the other way around. Interviews were conducted on Zoom or in person at the stable based on the interviewees’ preference and logistical considerations. Onsite interviews were conducted in a quiet room with no other participants, to ensure confidentiality and anonymity. All interviews were recorded and transcribed.

In Norway, the study included 7 representatives from national equine centers, stakeholders, and organizations, 11 representatives from riding schools, and 1 representative from a harness racing establishment. In Sweden, representatives from 10 riding schools and 1 harness racing establishment were interviewed, alongside 6 persons from national equine centers, stakeholders, and organizations.

At the national level, we contacted the national organizations in Sweden and Norway and asked them whether they could participate in interviews. For the selection of the Norwegian riding schools, we used a website in Norway called rideskoler.no [46] to select the interviewees. From their list, riding schools in the south of Norway were contacted because of their proximity to the Norwegian researcher. Another aspect of the criteria was size, and riding schools with 15–30 horses of medium size or more than 30 horses of large size were contacted and asked if they wanted to participate in the study. Twenty-five riding schools in Norway were contacted, and ten responded (five from medium sized and five from large riding schools) and participated in the study. In Sweden, the selection of riding schools was based on membership of the Swedish Equestrian Federation (SEF), size, and location. In 2022, there were 841 riding clubs which were members of the SEF, and 450 of these were riding schools [44]. We created an Excel document for these and sorted them according to their addresses. Thereafter, the website Hästnäringen i siffror [1] was used to sort the riding schools according to number of members and horses. Like the Norwegian case, we created three groups: large, medium-sized, and small riding schools, and contacted 30 riding schools. We received answers from nine big riding schools, whereas only one of the medium sized answered and participated, and zero from the small riding schools answered, even though they were reminded several times.

General questions about the riding center were asked, such as, ‘Can you tell me about the riding center and the work you are engaged in?’. Questions related to the environmental work at the stable included queries such as ‘How do you perceive the environmental challenges within the equine sector?’, ‘Does the riding school have any environmental or sustainability policies?’, and ‘Does the organization allocate resources—financial, personnel, knowledge, and expertise—for environmental initiatives?’. These questions were followed by inquiries about the future, such as ‘What do you hope the equine sector will look like in 10 or 50 years?’ and ‘How should a sustainable equine sector look in the future?’.

### 4.3. Analysis

As we used a mixed-methods parallel design, step one was collecting the quantitative and qualitative data. Step two included analyzing them separately. For the survey answers, we conducted a descriptive analysis to gain a broader understanding of our participants. Subsequently, we utilized Excel to examine the percentages representing the extent to which each category was perceived as an environmental problem within each group. In addition, we did a correlation analysis in JASP to examine potential similarities and differences in attitudes related to age, nationality, and relation to the equine sector.

For the qualitative data, we employed a combination of deductive content analysis and inductive thematic analysis to identify patterns and themes [47,48]. The interview transcriptions were reviewed multiple times so that the researchers became familiar with the data and gained a comprehensive understanding of participants’ perspectives. Initial coding was conducted using a deductive analysis, wherein everything related to environmental sustainability and the participants’ attitudes was highlighted by different researchers. The deductive content analysis allowed us to structure our data according to pre-existing concepts and theories. Afterwards, an inductive thematic analysis was conducted and codes were organized into different themes; similar codes related to, for example, litter and transportation, were grouped together. If some of the initial codes did not fit into any themes or were mentioned only once, they were excluded from the data. The inductive approach allowed us to identify themes directly from the data, providing a rich, detailed account of the participants’ perspectives. The themes were further refined and defined, involving revisiting the data to ensure coherence and representativeness of the initial themes. The final set of themes was agreed upon by the research team representing the most meaningful patterns around attitudes towards sustainable change in the stable.

In step three, we compared the quantitative data analysis with the qualitative data analysis and related them to each other. We discussed convergencies and divergencies. Thereafter, we drew overall conclusions and constructed main themes (and subthemes) reflecting survey answers and interviewees’ responses in relation to pro-environmental challenges to and facilitators of transformation. The combined approach of deductive content analysis and inductive thematic analysis provided a comprehensive understanding of the data, ensuring a robust and nuanced interpretation of the participants’ views on environmental sustainability within the equine sector.

The following themes were constructed and will be discussed in the findings section of the article:All work for a better environment is important, but it is difficult for me to influence work in the stable.The greatest environmental changes, with four subthemes:
Taking care of silage plastic and keeping things tidy.I can stop driving…but then I’d rather stop riding.There is much that could be changed regarding energy consumption.They throw out a lot of bedding.
What can lead to change.

The last step of the mixed-methods parallel design involved interpretation. The interpretation in this study was supported by comparing our results to previous research and by revisiting the COM-B model. We strived to understand how answers in the survey and interviews could be interpreted in relation to the three components of the COM-B model: capabilities, opportunities, and motivation. How answers in the survey and interviews could be interpreted were discussed in the research team and examples that illustrate the different components were agreed on. The interpretations will be presented together with a discussion on how behavior affects the equine sector and can be changed in the last section of the article.

## 5. Results

### 5.1. All Work for a Better Environment Is Important

The first theme, ‘All work for a better environment is important, but it is difficult for me to influence work in the stable’, underlines that environmental sustainability is important to those who responded to the survey. A comparison between our results and Svala’s from 2008 [35] suggests that there has been a (possibly ongoing) transformation regarding the equine sector’s attitude towards environmental sustainability. In the survey, almost all, 92%, believe that environmental sustainability is an important issue for the sector, and 70% would like to learn more about how environmental work can be undertaken in the stable. A majority, 69%, answered ‘yes’ to the question of whether they wanted to be involved in influencing environmental sustainability work in the stable, 6% said they did not want to, and 25% said they did not know if they wanted to.

The respondents to the survey also called for more influential environmental policies. Table 2 shows that 24% of respondents do not know if the stable has any policies, while others mention the absence of policies (55%).

The respondents gave many examples of how environmental work could be further developed at both structural and individual levels. Many expressed that they have both the motivation and interest to engage in environmental sustainability in the stable but find it challenging to initiate action and influence others. Just over half believed they have little or no opportunity to influence environmental work in their stable. They expressed it as follows in the free text answers in the survey:

A lot of the work with environmental sustainability in the stable is the responsibility of the stable owner. Manure management, choice of forage, choice of bedding, optimization of energy usage, installing solar panels, etcetera.

The engagement in environmental sustainability is also evident in the interviews; however, it is more complex. Both the respondents to the survey and the interviewed ownership of the stable define responsibility for actions related to pro-environmental changes. The interviewees expressed that a pro-environmental transformation is important, but that the work is difficult as decisions are taken elsewhere.

We have talked, the environmental group, about solar cells. They (roofs of the riding arenas) are perfect roofs for solar cells. But with this subletting, it gets complicated. … it is also an ambition in the municipality that all municipal facilities should have it. (even so) they just laugh, these officials with whom I have a dialogue. We also have our company-owned riding arena, so we thought we could put it (solar panels) on that. But then you were not allowed to convey energy between two properties. (Interview 10, Sweden)

In the quote, the interviewee refers to the complicated owner structure of many riding schools. Some are owned as private companies, while others are owned and run by voluntary organizations or the municipality. For many, the ownership is mixed [49] and therefore responsibility is redirected, in accordance with Wormbs and Wolrath Söderberg [36]. Another identified obstacle is regulations related to conveying energy between properties. It is, however, notable that horse and human welfare (expressed in relation to safety in Svala’s study [35]) was not pointed out as an obstacle to change.

### 5.2. The Greatest Environmental Challenges

The second theme draws from responses in the survey and interviews related to what the respondents to the survey and interviewees identified as the most important to prioritize to improve the environment in society and what are the biggest environmental challenges in the stable (see Figure 1 and Figure 2). This allowed us to see if respondents’ and interviewees’ environmental engagement in society, in general, is reflected in their environmental engagement related to the stable.

The responses were similar irrespective of the respondents’ role in the equine sector (Figure 2). Respondents to the survey believed that littering is the most significant environmental problem. There are, however, some differences. With regard to transportation, those with horses as a hobby consider this as more of a significant environmental issue than those working in the sector. The opposite applies for energy consumption and manure management; regarding these issues, those working in the sector see them as more important than the hobby riders.

Respondents to the survey also had the opportunity to explain what they saw as the most significant environmental challenge in society and in the stable, in the free text answers. In these, a majority highlighted consumption. It is expressed as follows:

General overconsumption, just like in all other sectors of society, this is a significant problem. Prices for riding equipment, such as blankets, are the same as in the 1980s, leading to grotesque overconsumption of items. Fashionable colors on bandages and saddle pads, glitter, and nonsense are bought like never before. (Survey)

This respondent claims that today’s prices are the same as prices in the 1980s and that this, together with fashion, spurs consumption. Problems with textile consumption are pointed out specifically (consumption of leather is not mentioned in the same way). In the interviews, textile consumption is talked about as a problem too. Here, we also find suggestions of solutions related to second-hand markets, ideas of mending and producing new products out of old ones. One of the interviewees said,

The riding school has a sewing room with a fantastic machine. They make blankets, mend them, and save little bits and little hooks and all sorts of things. They also fix halters and bridles. When the blankets cannot be fixed, they sew bags out of them. They are very active in trying to use everything. (Interview 1, Sweden)

Although not commented on by the interviewees, equestrians in other studies point to the importance of social media in their everyday life around the horse [50]. It is likely that equestrians’ consumption patterns are affected by social media influencers in the equine sector advertising products claiming that these will improve horse welfare. How consumption (or not consuming) and horse welfare are related to each other was, however, not problematized by the interviewees. They mention that the horse does not really need blankets of different colors, but they also underline that it is difficult to tell others what to buy and not.

#### 5.2.1. Take Care of Silage Plastic and Keep Things Tidy

Within the second theme, ‘the greatest environmental challenges’, we have constructed four subthemes. The first subtheme is ‘taking care of silage plastic and keeping things tidy’. A majority of respondents to the survey, 55% (264 individuals), perceive ‘littering’ as one of the most significant environmental challenges in the stable. Respondents aged 30 or younger believe that littering is a more significant problem than those older than 30 (see Figure 3) and Swedish respondents are more prone to identify littering as a problem (see Figure 4).

It is not entirely clear how to interpret what the respondents include in ‘littering’. Concerning the environmental issue and society, littering ranks fourth according to the respondents (Figure 1). Does this mean that stables have a higher degree of littering than the rest of society? Some insights can be obtained from the free text responses, where, for example, silage plastic is described as something found everywhere around horse farms, both on the ground and inside horse stalls and yards. Respondents also indicate a need for better waste management, including setting up waste sorting; recycling plastic packaging ‘instead of burning it’; separating plastic, paper, and food waste into different containers; and ‘ensuring that the silage area is handled properly’. The interviews help us to understand the issue further. After discussing the problems, the interviewees offered solutions.

We have improved waste sorting. For example, there are many types of waste. We sort (the waste into the following categories) agricultural plastic, paper, general waste, and horseshoes (the metal is in a separate container). We had to make an agreement with a different waste management company than the municipality offered, because we need a lot of different types. (Interview 2, Norway)

#### 5.2.2. I Can Stop Transporting… but Then I’d Rather Stop Riding

A second subtheme, I can stop driving…but then I’d rather stop riding, refers to transportation of many different kinds. A majority of survey respondents believe that carbon emissions from travel and transport are one of the most significant environmental challenges for society. According to 55% of the respondents, transportation is also seen as one of the most significant problems facing the equine sector. In this study, transportation is used as a ‘catch all’ term that includes transporting feed and bedding to the stables, taking manure and waste away from them, and travels for employees, students, and members getting to and from the stables. It can also involve transporting horses for training and competition or to the veterinarian or farrier. The question of transportation is complex, and it is not always easy to change one’s way of meeting transportation needs.

Many survey respondents (67%) use a car to get to the stable, and only 10% of these people have an electric car (more in Norway). Some mention the lack of public transportation, the difficulty of cycling without bike lanes, or the stables being located in rural areas as reasons for driving. It is interesting to note that even though horse welfare is used as an argument for some ways of treating the horse and is questionable in relation to transporting horses for training and competition, it is not problematized to a great extent in the answers in our study. In the free text responses, one respondent wrote,

I can stop transporting horses to the indoor arena and instead navigate through dangerous roads among tractors and trucks. But then I’d rather stop riding. (Survey)

There were some differences between groups within the survey sample. The Swedish respondents see transportation as a bigger challenge than Norwegian respondents (see Figure 4). There are also differences related to age, as it is the group above 30 years who perceive transportation as a significant problem (see Figure 3). In both Sweden and Norway, hobby riders (rather than those working in the sector) believe that transportation is a significant environmental challenge in equestrian sports (see Figure 5).

Why we see differences in the answers in relation to position in the equine sector is difficult to explain. A possible answer could be that hobby riders are more pre-occupied with how they travel to and from the stable and their dependence on cars, whereas stable managers offer solutions related to the fact that fodder and bedding could be transported fewer times. In the interviews, the location of riding schools is problematized:

Few facilities are located in the (town) center. They are located outside of town. …we should try to carpool so that not everyone comes with their own car, and we will work to ensure that the municipality can offer a good bus route. Because the buses stop running already at 18:00 in the evening and by then we have barely started, there are quite a few lessons left then. So even if the young people get here by bus, they don’t get home by bus. So, I think, there is a lot to do there. (Interview 10, Sweden)

Many, 41%, of survey respondents would like to change their travel behaviors to make them more environmentally friendly. It was reported that this might come from having more time, resources, knowledge, and energy. The thinking structure relates to what Wormbs and Wolrath Söderberg call ‘the limits of reality’ [36]. But there are also those who discuss various collaborative solutions to transportation issues (like the interviewee quoted above). This can involve collaboration between riders to carpool and government entities controlling public transport and road networks to make it easier to use public transport or cycle.

Other solutions related to transport were that those who have their horses in the same stable could buy feed from the same supplier. Almost half of the survey respondents (48%) state that they use locally produced feed to reduce transport emissions, and more than half (56%) wish to use locally produced feed. Beyond issues relating directly to transportation, the issue of feed concerns the importance of choosing the right supplier, for reasons related to horse welfare. We define this as ‘other values…’ in accordance with Wormbs and Wolrath Söderberg [36]. It is not certain that locally produced feed meets the quality one desires, and sometimes it is challenging to determine if the feed is good enough. One respondent to the survey expressed.

If forage producers became more aware of the importance of feed analysis, it would be easier for us to choose more locally produced and preferably organic feed. Since we have several horses with health issues (EMS, PPID, laminitis), it is a must to have forage low in sugar/ESC values, and when choosing a forage supplier, many options are eliminated due to the lack of analysis values. (Survey)

The issue of horse welfare can thus be a considerable obstacle to increasing pro-environmental behaviors in some regards, as expressed by one of the interviewees.

Food safety and horse welfare are probably stronger than environmental concerns at the moment when deciding how to build labor-saving and horse welfare. I think that stands very, very high. (Interview 5, Norway)

This is, however, also related to ‘limits of the reality’ connected to the fact that the rationalized agriculture does not offer a biodiverse forage that may be better to ensure horse welfare [51].

#### 5.2.3. There Is Much That Could Be Changed Regarding Energy Consumption

Respondents to the survey highlighted how energy consumption is one of the significant environmental challenges for both society (59%) and stables (44%). Subtheme three, ‘there is much that could be changed regarding energy consumption’, relates to the wake of increased electricity prices in Sweden and Norway in 2022 and 2023. Here, 43% of respondents stated that they try to reduce energy consumption, and as many as 51% would like to be able to influence the stable’s energy consumption. There are differences between the various groups within the survey sample. Age correlates with the view that energy consumption is a problem, with older individuals emphasizing this more than younger ones (see Figure 3). Norwegian respondents see energy consumption as a less significant issue than the Swedish ones (see Figure 4). Whether this can be explained by somewhat lower prices is difficult to say. In relation to position in the sector, it is more difficult to draw conclusions (see Figure 6). Energy consumption seems to be equally important for hobby riders and those working in the sector, except for the category involving both these groups.

Respondents to the survey and interviewees also have thoughts on what could be done to reduce energy consumption. The answers include lighting (changes to LED lights) and the difficulties with heat leakages when warming up different areas in the stable:

…there is much that could be changed. Self-closing doors to the tack room, especially in winter. Lighting with motion detectors in selected places, for example, toilets. (Survey)

When it comes to all the buildings, there have been some individual measures taken. It can certainly be improved, but yes, there have been changes to LED lights, and we can regulate the heat a bit up and down, and such things to save electricity. (Interview 2, Norway)

Problems relating to the facilities are raised too. Some of the interviewees underline that their stables are old buildings and that changes are difficult for economic reasons (‘limits of reality’). Again, it is interesting to note that horse welfare is not used as an argument against change. The interviewees express that they hope that they will get support (from municipalities) to renovate the facilities and that the renovations will include solutions for reducing energy consumption. However, they do not offer new solutions based on letting the horses be outside for longer hours (which could be seen as a horse welfare and pro-environmental change). There are also answers related to the fact that energy consumption needs to be registered in relation to the many different issues involved:

(We need to) …conduct an energy audit at the riding school to identify our largest energy consumers and then implement measures such as changing lighting, adjusting temperatures in certain parts of the stable, changing routines, and so on. (Survey)

#### 5.2.4. They Throw Out a Lot of Bedding

The fourth subtheme, ‘they throw out a lot of bedding’, relates to manure management as one of the most important environmental challenges for the sector. Surprisingly, only 22% of respondents in the survey believe that manure management is one of the most pressing environmental issues, and 17% mention manure management as something they would change if they had more time, knowledge, or energy. Of all those who responded, 22% state that they are actively doing something to improve manure management practices, while almost a third (29%) state that work on improving manure management is already underway in their stable. In the free text responses, several respondents wrote about how practices can be improved but that a change can be very difficult:

(Taking action) would be to muck out the paddock. But it’s 3 hectares, used by three horses, not close to water. So, the environmental impact is small, but it would mean a very large effort from me. Additionally, it is sometimes used by cows, whose owners do not muck out. From that perspective, the job of mucking out after my horses feels a bit as it is worth nothing. (Survey)

The answer relates to the thinking structure ‘the limits of reality’ [36]. Others suggest that mucking out practices could be improved by ensuring that less bedding is thrown out and that there is a norm prescribing using too much bedding. In the quote below this is ascribed to ‘others’:

They throw out a lot of bedding, and the norm is freshly bedded boxes every day. (Survey)

In the interviews, the issue of manure management is raised too. Here, it is problematized in relation to who is responsible for taking care of the manure and transporting it. The challenges relate to what kind of bedding is used, whether the municipalities offer a solution, and the possibility of developing thermal power plants using manure. In addition, the work environment of employees in the stable is brought up:

What is also still a challenge for us is being able to keep the paddocks mucked out. We have an Achilles heel, and we can’t muck out the paddocks as much as I would like. We have 70 horses to muck out in the stable and in the paddocks. That requires a lot of time, commitment, and money. (Interview 2, Sweden)

The mucking out is heavy work and needs to be done often—whether it is in the stable or in the paddocks, and stable work is seldom mechanized. Therefore, stable managers depend on employees to perform heavy work. Giving them extra chores or asking them to muck out big paddocks or paddocks far from the stable is difficult given the expenditure (‘limits of reality’).

### 5.3. What Can Lead to Change…

The third main theme, what can lead to change, is constructed in relation to respondents’ and interviewees’ answers on what could make them change their horse related behaviors in a more sustainable direction. A majority of the survey respondents, 61%, believe that economic incentives could change their behaviors. Horse welfare is also a crucial motivator for 55% of respondents. Knowledge is the third factor considered (by 45%) as essential for changed behavior. The economy is an important challenge too, as revealed in the interviews:

It’s very difficult for the stable because we don’t earn much money. We don’t make money; you can say that we break even. (Interview 6, Norway)

It’s a matter of cost, I’d say. Because it is a non-profit association. All such (environmental/green) investments cost money. (Interview 4, Sweden)

As we have already discussed above, it is evident that the importance of horse welfare for the respondents of the survey and the interviewees is seen in answers related to what can lead to a pro-environmental transformation of the equine sector. Often, it is expressed as an answer to why something is difficult to change, such as in the following quote:

It’s the small things, that we have the light turned on for quite a long time in the stable. Both in the mornings and in the evenings. Because the horses feel better when they get the light. They get much finer fur. I see that they feel better with the lights on. But, from an economic and environmental point of view, we should turn off the turned on, when we are not there ourselves. So, there are probably a few things like that, where we choose to prioritize what is best for the horse. (Interview 1, Sweden)

In relation to knowledge, there is a wide range of answers, from arguments that more knowledge is not needed, to arguments that the equine sector needs to improve its knowledge and competence in relation to environmental issues. The knowledge-gap is related to how well-educated the staff are in the stables. With better education, more could be done.

## 6. Concluding Discussion and Suggestions for Practice

The results of this study can be summarized by stating that since Svala’s study was published 2008 [35], the equine sector has developed a stronger environmental commitment. Many in the sector consider environmental issues important—at least among those who responded to our survey and participated in the interviews. At the same time, many feel that they lack opportunities to influence the kind of transformation they believe is needed. The factors noted by other researchers as being challenging for the broader equine community to address in their quest to embrace a more sustainable future are also found in our responses [6,7], but somewhat surprisingly, manure management is not considered a significant problem and it is presented as a solution (for example, through manure being used for creating energy). Perhaps respondents to the survey believe that manure management is already regulated through the municipality’s environmental management. Instead, littering, transportation, energy consumption, and feed production are highlighted as primary concerns.

Previous studies point to the importance of understanding how people reason and argue in relation to pro-environmental transformation to accomplish change. For sports participants, arguments relate to perceptions of fair play in connection to competition, whereas people in outdoor activities are more prone to underline the importance of saving nature [21,23,24]. These arguments are not found in our study. Some of the thinking structures on climate delay presented by Wormbs and Wolrath Söderberg [36] are more in line with our findings. However, they do not present arguments related to a climate account. Instead, our respondents and interviewees reason around ‘the limits of reality’. They say it is too difficult for them to change their transportation patterns as they need the car to be able to do activities with horses: to transport themselves, fodder, bedding, and horses. Public transport does not exist, would take too long, and is not suitable for what they need. Another argument evolves around ‘redirecting responsibility’. In our study, the respondents and interviewees have identified that they are responsible, but simultaneously they experience that they are not able to influence how the transformation should be done. They also point to what ‘others’ do (for example, throw out too much bedding). In that way, they redirect the responsibility for pro-environmental change to stable owners, boards of the riding schools, or municipalities. The final set of arguments relates to what Wormbs and Wolrath Söderberg [36] identify as ‘Other values and social relations’. They show that their respondents refer to their own health and well-being in their arguments for why they do not change their behavior. In our study, horse welfare (rather than human well-being) is central. This argument is crucial for the sector, and we suggest that it can prompt individuals in the sector to change their behavior. However, it is important to underline that horse welfare is also used as an obstacle to change and sometimes is not referred to (in relation to for example consumption, transport, and energy consumption). It also points to the importance of considering that arguments and practices are related to specific contexts and that environmental commitment in one context (society at large) may not necessarily transfer to another context (the stable) [52,53]. Knowledge and economics are also considered important for our respondents and interviewees. Like Tjønndal et al.’s [22] study, which indicates that social and environmental sustainability goals are woven together in the projects they investigate, our Norwegian interviewees discuss social and environmental sustainability together. In the Swedish material, however, we do not find a corresponding intertwinement in the responses. This is possibly linked to the fact that the Swedish equine sector has come further in their work with environmental sustainability (an explanation offered by the Norwegian interviewees).

As a closing discussion, we return to the model for change that we initially presented. An interpretation of the responses points to the fact that the capability for transformation exists, there are skills for change (even though some emphasize how challenging tasks like mucking out paddocks can be), and there is knowledge (although some individuals wish for more). In relation to the opportunity component, there seem to be higher barriers to change. In particular, physical opportunities such as location, resources, and time are difficult for individual actors to influence. The discussion on energy production and consumption offers an interesting example. For some of the riding schools, solar panels are presented as a pro-environmental solution. There are, however, some regulations on how the produced energy can be distributed between properties which becomes an obstacle. Another example is connected to manure management. Manure could be used in cogeneration plants, but these are usually not built for riding schools. Investments are needed and economic factors are also identified as barriers to change. Pro-environmental investments are perceived as expensive, and many stables are already financially stretched, with low levels of perceived economic viability. It is likely that collaborative partners (aka social opportunities) are crucial for change, but these opportunities vary greatly according to location. Riding schools owned by local governments should be able to find suitable partners to invest in solar panels and developing water collection solutions. Associations also have the possibility to seek funding for pro-environmental projects through the National Sports Federation [54].

Social opportunities also involve cultural norms. Over-consumption was mentioned by the respondents and interviewees as a challenge. It is likely that consumption is spurred by cultural norms and these norms need to be questioned and changed for the sector to transform. More research on how horse owners and riders perceive horse welfare and consumption is needed to problematize these norms. Above, we showed that littering is considered one of the significant problems for the sector. It is not easy to explain why littering is seen as an important issue. Stables and stable yards do not stand out as littered places compared to cities or roadsides. However, littering could imply a kind of general disorder that the respondents are unhappy with. In the shared space of a stable, everyone is expected to keep their belongings in order. The cultural norm includes having ‘a place for everything’; properly taking care of saddles, bridles, and blankets; and sweeping and tidying-up outdoor parts of the facility. Order is often motivated by safety (horse and human welfare) but is heavily influenced by cultural norms regarding how things ‘should look’ in a traditional stable culture [55,56,57]. Even though this orderliness does not primarily concern addressing ecological challenges, it might be something to build on in terms of how more pro-environmental actions can be embedded into cultural expectations of what is ‘right’. There is already an essential prerequisite here concerning the shared responsibility for the environment, and this could be used for instilling changes in practices related to waste sorting, manure management, and recycling.

The COM-B model’s remaining component is motivation. We interpret answers from respondents and interviewees as pointing to the fact that they are highly motivated to make changes in a more sustainable direction. To be able to depict in what ways motivation in relation to different changes can be identified as automatic or reflective, a more in-depth investigation would be needed. However, arguments such as ‘this is how we do things here’ (automatic motivation) that can be linked to a traditional stable culture mentioned above are present. We suggest that this automatic motivation needs to be made conscious for people’s behaviors to change, even if it seems likely that some change could be spurred by this automatic motivation, e.g., ‘in the stable, we keep things tidy’. The ‘keeping tidy’ could include sorting waste and mucking paddocks.

Motivation to act in more pro-environmental ways must and can be related to horse welfare. Indeed, a recent study in occupational science shows that the motivation for horse welfare can be so high that those working in the stables prioritize the health and welfare of the horses over their own [58]. This would mean that if ecological sustainability were to focus more deliberately on improving horse welfare, a large obstacle to change may be removed. There are, however, some obstacles that will be more difficult to overcome, for example: in what way can horse welfare be used as an argument (that equestrians will accept) to decrease transport and consumption? Challenges that are more easily solved are connected to forage, manure, and bedding. If the horses benefit from having higher quality feed that can be found closer to the stable, managers of stables and horse owners would choose this feed (if it is not too expensive). Another example relates to the mucking out of paddocks. If the mucking out is presented as a way of reducing the parasite pressure, it would be easier to convince individuals in the sector (than using nutrient leakage as an argument). A third example relates to blankets. If horse owners and riders can be convinced that blankets (in many different colors) are not needed for many horses, consumption patterns may change. A fourth example concerns bedding. If it is shown that it is healthier for horses to stand on a certain bedding that is more sustainably produced, then motivation to adopt more pro-environmental practices becomes a natural by-product of increasing horse welfare. In this way, a pro-environmental transformation of the equine sector goes hand in hand with ensuring and improving horse welfare for a sustainable future.

## Figures and Tables

**Figure 1 animals-14-00915-f001:**
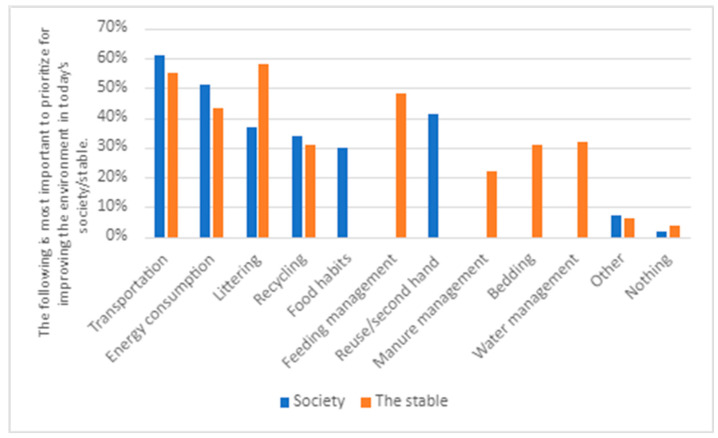
Responses to the question ‘I believe the following is most important to prioritize for improving the environment in today’s society’ and ‘I believe the following are the biggest environmental challenges in the stable’ (N = 685).

**Figure 2 animals-14-00915-f002:**
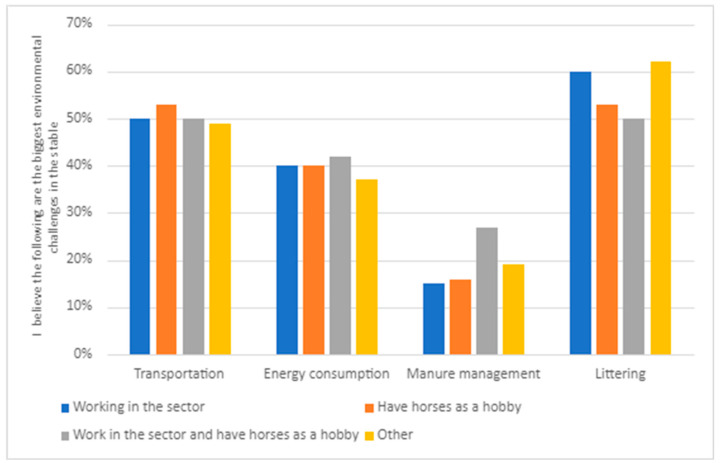
Responses to the question ‘I believe the following are the biggest environmental challenges in the stable’, divided into groups within the sector (N = 534).

**Figure 3 animals-14-00915-f003:**
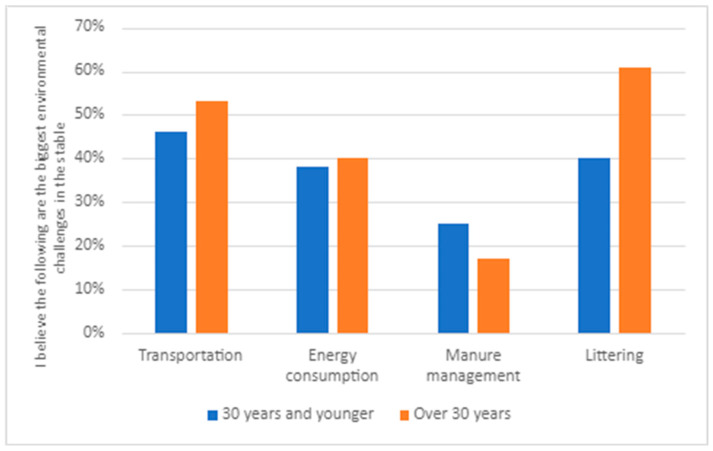
Responses to the question ‘I believe the following are the biggest environmental challenges in the stable’ divided into 30 or younger and those older than 30 years (All respondents N = 530; 30 years or younger N = 204; older than 30 years N = 326).

**Figure 4 animals-14-00915-f004:**
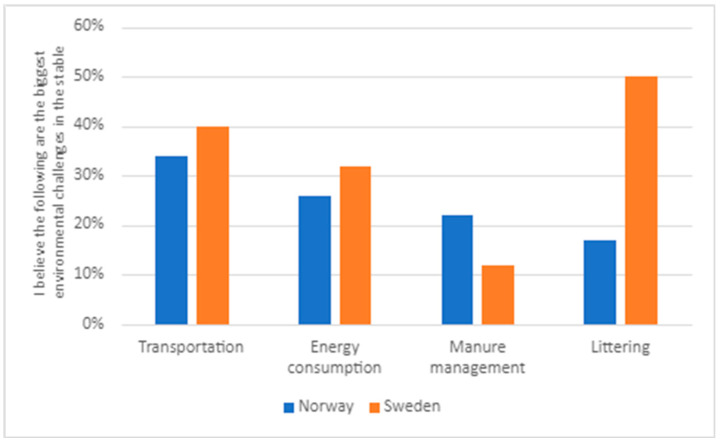
Responses to the question ‘I believe the following are the biggest environmental challenges in the stable’ divided into Sweden and Norway. (All respondents N = 534; Norwegian respondents N = 242; Swedish respondents N = 291).

**Figure 5 animals-14-00915-f005:**
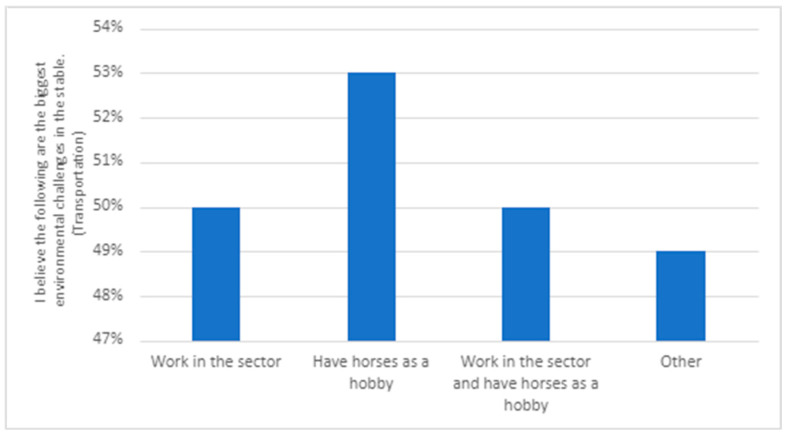
Responses to the question ‘I believe the following are the biggest environmental challenges in the stable’ regarding transportation and divided by the respondents’ relationship to the sector. (N = 534).

**Figure 6 animals-14-00915-f006:**
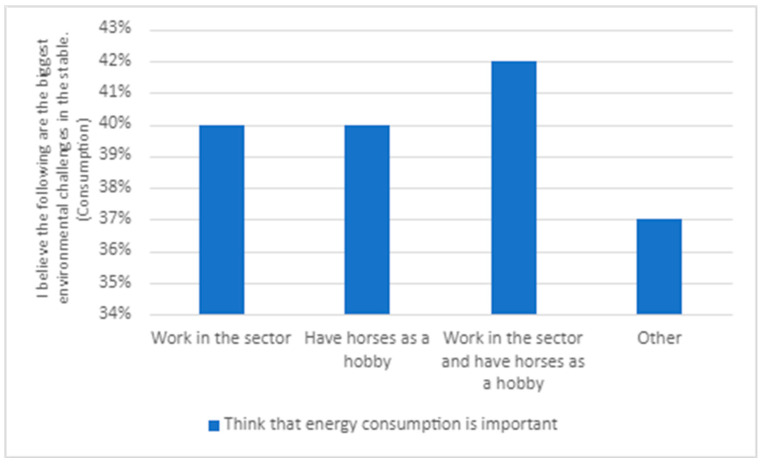
Responses to the question ‘I believe the following are the biggest environmental challenges in the stable’.

**Table 1 animals-14-00915-t001:** Survey population.

	Sweden	Norway	Percentage (%) of the Total Population	Total
Mean Age (standard deviation)	41 (±15 years)	29 (±14.1 years)	NA	NA
Number of years in the equine sector	29 (±15.2 years)	19 (±13 years)	NA	NA
Number of individuals with horses as a hobby	173	57	43	230
Number of individuals working in the sector	30	10	7	40
Number of individuals working in the sector and having horses as a hobby	75	90	31	165
Others	13	85	19	99
Total	291	242	100	534

NA = not applicable.

**Table 2 animals-14-00915-t002:** Respondents’ perception of whether the stables where they are active have an environmental policy or not (percentage in parentheses).

The Stable Has An Environmental Policy	Sweden	Norway	Total
Yes	67 (21)	24 (18)	91 (21)
No	180 (57)	65 (49)	245 (55)
Don’t know	66 (21)	43 (32)	109 (24)
Total	313 (99)	132 (99)	445 (100)

## Data Availability

The data presented in this study are available on request from the corresponding author. The documents used can be found using our references and transcriptions from the interviews are saved at the Malmö University server and not published OA because of ethical reasons related to protecting the anonymity of our interviewees.

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
