# Peer review of "Pro-Environmental Transformation of the Equine Sector—Facilitators and Challenges"

_animals, 2024, doi:10.3390/ani14060915_

Round 1

Reviewer 1 Report

Comments and Suggestions for Authors

Really pleasing to see research of this nature being undertaken to aid wider understanding of the challenges for contemporary equestrians on multiple levels and to ensure a sustainable future; important to understand and very interesting to review.

I really like your work and overall arguments are well presented but I feel there is scope to revisit and rejig to increase the impact of the content to the sector.

Simple summary: Clear overview provided

Abstract: comprehensive review of key points, may be useful to add in margin of error detail to show survey response rate is representative of Swedish and Norwegian equestrians

Introduction

Nice intro provided, provides relevant background and sets scene well, scope to link use of COM-B model to recent application of this in horse welfare as an additional example here (Wolframm, I.A., Douglas, J. and Pearson, G., 2023. Changing hearts and minds in the equestrian world one behaviour at a time. Animals13(4), p.748.)

Line 62: could add an example here perhaps

Line 111-2: may be useful addition to align need to consider horse welfare as a cornerstone of SLO and sustainable equestrian sector in the largest sense, could align to FEI Equine Ethics and Wellbeing Commission for example

Line 142: your design is more mixed methods as you are combining survey and interviews

Methods and Data

I think reorganising to subdivide methods from initial data results would improve the readability of the article

Would be beneficial to include copy of the survey as a supplementary file

Including a sample size calculation here (or margin of error in results) to demonstrate how representative your numbers are to the equestrian population in the countries included would be beneficial

For the survey and interviews, further detail as to inclusion and exclusion criteria for participants would be useful to include and please clarify if these occurred simultaneously or sequentially

For the interviews, details of the experience of the interviewer, environment, how these were recorded and what data were evaluated would be a useful addition

Further details re the approach used to inform thematic analysis is needed as it reads currently like deductive content analysis was applied – linking to the thematic framework and how coding was applied and themes were triangulated would help alleviate this impression

Line 286: please outline how the COM-B model was applied in more detail and how this informed revisiting the survey and interview data

Results

Personally, I would prefer to review survey results then then interviews with application to COM-B and survey interrelated in the latter – the current format could work but think need to set up more in methods first and watch the interview results do not overtake the content here. I am not fully sure how the COM-B model has been applied in here and wonder if this could be made more explicit for the reader, perhaps by aligning the content to capability, opportunity and motivation (see EEWC approach and Wolframm paper above for examples for how this could potentially be undertaken) – maybe remove from here and expand upon in discussion, expanding existing discussion and bringing together in a visual context too?

Liked the application to Svala’s work to get a feeling of context over time

Some very interesting perspectives raised.

Diagrams should be figures and please include axes labels

Discussion

I feel there are some missed opportunities here to really showcase your work. See comment above re application of COM-B to the results and suggestions for how to optimise these and increase impact (and maybe application) of your work more.

Comments on the Quality of English Language

Good

Author Response

Thank you so much for your valid review!
We attach a document where we have answered your questions step by step.

Reviewer 2 Report

Comments and Suggestions for Authors

This is a very interesting paper.

There are minor changes to the language and phrasing needed throughout.  I have provided a copy of the submitted manuscript with 51 comments throughout - mostly editorial.

I have two important concerns:

1. The authors state that ethical approval was not required - however in my experience across 3 different institutions conducting research human ethics approval is required for both surveys and interviews.  (I am satisfied however that consent was obtained appropriately.)

2. The method section needs to be restructured to make clear the two different parts of the data collection, namely the survey and then the interviews.   These two methodologies, and associated terminologies, seem to be used interchangeably which makes understanding more difficult than it ought to be in places, and does not currently enable an independent researcher to replicate the study.  

Comments on the Quality of English Language

Generally acceptable but lapses lead to difficulties in grasping the key points of sentences/paragraphs. I have highlighted some areas that can be amended however the entire manuscript would benefit from a native English-speaker to look it over prior to resubmission. 

Author Response

(The authors gave the same response as above.)

Round 2

Reviewer 1 Report

Comments and Suggestions for Authors

Thank you for making the revisions to the manuscript, which I hope you agree has improved its impact. There are a couple of minor areas which require further clarification prior to publication:

Line 127: error SOL not SLO

Line 316: the method of analysis described for the interviews related to environmental sustainability aligns to conventional deductive content analysis rather than thematic analysis (which is usually inductive in its approach). The latter does also seem to have been conducted on the narrative data. Suggest revising and clarifying that you used both a deductive (content analysis) and inductive (thematic analysis) approach. 

Comments on the Quality of English Language

Proof read required to capture odd minor errors

Author Response

Thank you so much for your final comments!

We have worked through the text again and made adjustments according to your notes.

All the best,

Anna et al